# Django: Detecting Trojans in Object Detection Models via Gaussian Focus Calibration

**Guangyu Shen**[*]
Purdue University
West Lafayette, IN, 47907
shen447@purdue.edu

**Siyuan Cheng**[*]
Purdue University
West Lafayette, IN 47907
cheng535@purdue.edu

**Guanhong Tao**
Purdue University
West Lafayette, IN, 47907
taog@purdue.edu

**Kaiyuan Zhang**
Purdue University
West Lafayette, IN, 47907
zhan4057@purdue.edu

**Yingqi Liu**
Microsoft
Redmond, Washington 98052
yingqiliu@microsoft.com

**Shengwei An**
Purdue University
West Lafayette, IN, 47907
an93@purdue.edu

**Shiqing Ma**
University of Massachusetts at Amherst
Amherst, MA, 01003
shiqingma@umass.edu

**Xiangyu Zhang**
Purdue University
West Lafayette, IN, 47907
xyzhang@cs.purdue.edu

## Abstract

Object detection models are vulnerable to backdoor or trojan attacks, where an attacker can inject malicious triggers into the model, leading to altered behavior during inference. As a defense mechanism, trigger inversion leverages optimization to reverse-engineer triggers and identify compromised models. While existing trigger inversion methods assume that each instance from the support set is equally affected by the injected trigger, we observe that the poison effect can vary significantly across bounding boxes in object detection models due to its dense prediction nature, leading to an undesired optimization objective misalignment issue for existing trigger reverse-engineering methods. To address this challenge, we propose the first object detection backdoor detection framework Django (*Detecting Trojans in Object Detection Models via Gaussian Focus Calibration*). It leverages a dynamic Gaussian weighting scheme that prioritizes more vulnerable victim boxes and assigns appropriate coefficients to calibrate the optimization objective during trigger inversion. In addition, we combine Django with a novel label proposal pre-processing technique to enhance its efficiency. We evaluate Django on 3 object detection image datasets, 3 model architectures, and 2 types of attacks, with a total of 168 models. Our experimental results show that Django outperforms 6 state-of-the-art baselines, with up to 38% accuracy improvement and 10x reduced overhead. The code is available at https://github.com/PurduePAML/DJGO.

## 1 Introduction

Object detection is an extensively studied computer vision application that aims to identify multiple objects in a given image. It has been widely integrated into various real-world systems, including public surveillance [51], autonomous driving [14, 25], optical character recognition (OCR)[41], etc.

---

[*]Equal Contribution

37th Conference on Neural Information Processing Systems (NeurIPS 2023).

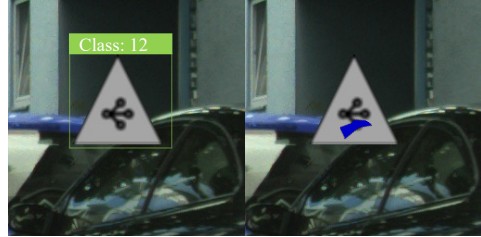 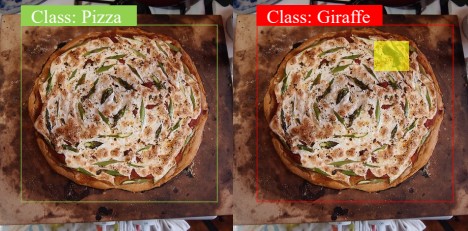

|  (a) Evasion trigger | (b) Misclassification trigger |

Figure 1: Triggers with different effects. The blue patch in Figure 1(a) is an evasion trigger, causing the bounding box to disappear. Figure 1(b) represents a misclassification trigger (yellow), leading to the misclassification of "pizza" as "giraffe".

State-of-the-art object detection systems, e.g., SSD [32] and YOLO [44], are primarily built on Deep Learning (DL) models [13], owing to their remarkable feature representation capabilities.

Despite the impressive performance of existing object detection approaches, they are vulnerable to backdoor attacks. Backdoor attack is a form of training-time attack, in which a malicious adversary strategically embeds an imperceptible *trigger* (e.g., a small image patch) onto a small set of training data. DL models trained on this poisoned dataset will predict the attacker-chosen *target label* whenever an input is stamped with the trigger. There are two major backdoor attacks in object detection, namely, *misclassification attack* [3] and *evasion attack*. As shown in Figure 1, misclassification attack causes the object detection model to misclassify an object as a target object, whereas evasion attack makes the model fail to recognize certain objects (i.e., classify traffic signs as the background). Such attacks can significantly hinder the deployment of object detection systems in real world as they may cause severe security issues or even life-threatening incidents.

To counter the backdoor threat, researchers have proposed a multitude of defense techniques aimed at safeguarding the model against backdoor attacks throughout its entire life cycle [59, 33, 69, 16, 26, 22, 18, 10, 54, 50, 36, 72, 27, 65, 31, 35]. Among them, trigger inversion [59, 55, 18] is one of the most popular backdoor scanning methods and has demonstrated its effectiveness in various tasks [59, 35]. Given a small set of clean samples and the subject model, trigger inversion techniques aims to determine if the model is backdoored or not by reverse-engineering the underlying triggers. The task of trigger inversion can be formulated as an optimization problem with specific constraints, such as trigger size [59, 18] or particular stylistic attributes [9, 42].

In order to detect backdoors in object detection, a direct solution is to adapt trigger inversion from the classification task to object detection. We however find that a simple adaptation of trigger inversion fails short. This attributes to the significantly denser output of object detection models due to the nature of the task. In specific, an object detection model, such as SSD [32], typically produces thousands of bounding boxes with corresponding class probabilities to capture objects of varying scales within a single image. Attackers can exploit this extensive output space to conceal injected triggers and mislead the optimization process during trigger inversion, causing the *loss misalignment issue*. That is, throughout the entire trigger inversion process, the loss values consistently remain either significantly high or extremely low, regardless of the *Attack Success Rate (ASR)* that denotes the ratio of misclassified samples when the trigger is applied. This misalignment property hinders the identification of high-quality triggers, leading to low backdoor detection performance.

Our study reveals that the underlying cause of the *loss misalignment issue* stems from the unequal impact of poisoning on individual bounding boxes within the backdoored model. More specifically, the backdoor trigger affects only a small fraction of bounding boxes, while leaving the majority of them largely unaffected. Consequently, the aggregated gradient information is obfuscated by remaining benign bounding boxes during inversion, thereby impeding the optimizer from accurately estimating the gradient direction. Based on this observation, we propose the first trigger inversion-based backdoor detection framework for object detection: DJANGO (*Detecting Trojans in Object Detection Models via Gaussian Focus Calibration*). The overview is shown in Figure 2. It features a novel *Gaussian Focus Loss* to calibrate the misaligned loss during inversion by dynamically assigning weights to individual boxes based on their vulnerability. Equipped with a label proposal pre-processor, DJANGO is able to quickly identify malicious victim-target labels and effectively invert the injected trigger lies in the backdoored model. Extensive evaluation of 3 object detection datasets, utilizing 3

different model architectures with a total of 168 models, demonstrates the superiority of DJANGO over six existing state-of-the-art backdoor scanning techniques. Specifically, we achieve up to 0.38 ROC improvement and a substantial reduction in scanning time.

## 2 Background & Threat Model

**Deep Learning based Object Detection Techniques.** Extensive research efforts have been devoted to deep learning (DL) object detection, encompassing various aspects such as neural network architecture design [32, 45, 2], training optimization [29, 61, 74], and robustness [71, 66, 6]. These endeavors contribute to a broad body of work aimed at advancing the field of DL object detection. Specifically, given an image $x \in \mathcal{X}$ that contains $p$ objects, an object detection neural network $g_\theta$ (parameterized by $\theta$) outputs $K$ ($K \gg p$) 5-tuples: $\hat{y} = g_\theta(x) = \{\underbrace{\hat{b}_k^x, \hat{b}_k^y, \hat{b}_k^h, \hat{b}_k^w}, \hat{p}_k\}_{k=1}^K = \{\hat{b}_k, \hat{p}_k\}_{k=1}^K$, shorten

Figure 2: Overview of DJANGO

as $\hat{b}_k(x)$ and $\hat{p}_k(x)$, which denote the center coordinates, height, width, and class probability of the $k$-th predicted bounding box. Note that $K$ is usually much larger than $p$ ($K = 8732$ in SSD [32], $p \approx 7.7$ in COCO dataset [30]). Box matching algorithms [32, 46] and *Non-Maximum Suppression*(NMS) [20] were introduced to address the challenge posed by the substantial box number discrepancy between the prediction and ground-truth in both training and inference stages.

**Backdoor Attack & Defense.** Backdoor attacks can be carried out through data poisoning with modified or clean labels [17, 34, 7, 58] and model parameter hijacking [47]. Initially, small image patches were used as backdoor triggers [17, 8, 48], and later on more complex triggers have been successfully injected in DL models [28, 64, 33, 37]. These attacks are not limited to image classification models; they also affect Large Language Models [70], code models [73, 52], Reinforcement Learning agents [60], pre-trained image encoders [24, 56], and object detectors [3, 39]. Researchers have proposed various approaches from defense perspectives. During the training stage, methods such as poisoned sample identification and filtering have been proposed [40, 53, 4, 57, 19, 43]. In the post-training stage, backdoor scanning techniques aim to recover injected triggers from trained models [59, 49, 54, 62, 10, 63, 35, 50, 15]. Backdoor removal techniques are employed to eliminate triggers with minimal resource consumption and performance impact. Real-time rejection of trigger-carrying samples during inference can be achieved using methods like [12, 16]. Furthermore, theoretical analyses have been conducted to understand backdoor attacks [67, 68, 72, 23]. In this work, we prioritize the practical application of trigger inversion-based backdoor detection, with a specific focus on object detection models.

## 3 Methodology

**Threat Model.** Our work considers the standard setting used in existing backdoor scanning literature [59, 33, 18, 10], where the defender has a small set of clean samples from each class in the validation set (10 in our paper), but no poison samples and white-box access to the model under scanning. The defense goal is to classify the benignity of the subject model. Our focus in this study is on *label specific* polygon triggers with two types of effects: *misclassification* and *evasion*, which cause the model to predict an object as a target object and the background, respectively.

### 3.1 Misalignment of CE Loss and ASR in Object Detection Model Trigger Inversion

**Backdoor Detection via Trigger Inversion.** Trigger inversion aims to decide whether the model is backdoored or not by reverse-engineering the injected trigger through optimization. Consider an image classification model $f_\theta$ parameterized by $\theta$ with $C$ classes and a small support set $\mathcal{S}_{c_v}$ from

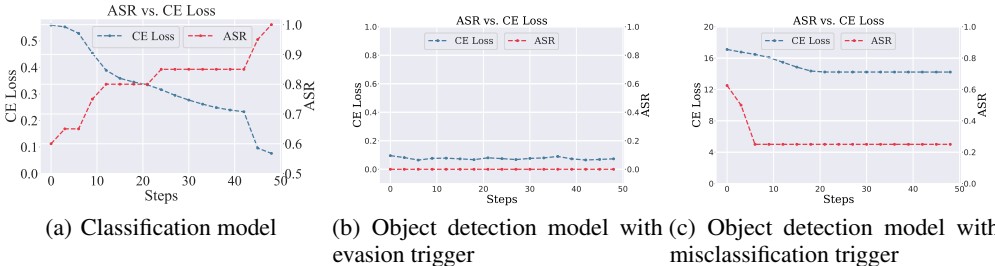

(a) Classification model    (b) Object detection model with evasion trigger    (c) Object detection model with misclassification trigger

Figure 3: ASR vs. CE loss during trigger inversion

each class $c_v$. Trigger inversion [49, 59] aims to solve the following optimization objective.

$$\min_{m,p} \mathbb{E}_{x \in \mathcal{S}_{c_v}} [\mathcal{L}(f_\theta(\phi(x)), c_t)] + \beta ||m||_1, \text{ where } \phi(x) = (1-m) \odot x + m \odot p. \qquad (1)$$

Variables $m$ and $p$ are the trigger mask and pattern respectively, to be optimized. The loss function $\mathcal{L}(\cdot)$ typically resues the task loss function as a measurement of the degree of the misbehaved model. In classification and object detection, it will be the cross entropy (CE) loss. $\beta$ is the coefficient on the regularization term, and $||m||_1$ represents the $\ell_1$ norm of the mask. Since the defender lacks the knowledge of the exact poisoned victim and target labels, the backdoor scanning requires optimizing Eq. 1 for every label pair $(c_v, c_j)$. If any of these pairs yields a trigger with an exceptionally small size, it is indicative that the model is trojaned.

Intuitively, it shall be effortless to adapt existing trigger inversion methods from image classification to object detection. Considering the disparity of these two tasks, object detection models generate much denser output for each input, i.e., thousands of bounding boxes versus one label in classification. We can make adjustments to Eq. 1 to optimize a trigger that can impact samples at the box level rather than the entire image. Given a sample containing a victim object, one naive modification is to optimize a trigger that can influence all bounding boxes that are close to the victim object. The objective function can be reformulated as follows:

$$\min_{m,p} \mathbb{E}_{x \in \mathcal{S}_{c_v}} \mathbb{E}_{\hat{o} \in \mathcal{T}_x} [\mathcal{L}(\hat{o}, c_t)] + \beta ||m||_1,$$
$$\text{where } \mathcal{T}_x = \{\hat{p}_k(\phi(x)) \mid \mathcal{J}[\hat{b}_k(\phi(x)), b^*] \geq \alpha\} \text{ and } \phi(x) = (1-m) \odot x + m \odot p. \qquad (2)$$

We denote the predicted probability and coordinates of the $k$-th bounding box of sample $\phi(x)$ as $\hat{p}_k(\phi(x))$ and $\hat{b}_k(\phi(x))$, respectively. The function $\mathcal{J}(\cdot)$ refers to the Jaccard Overlap, also known as Intersection over Union. $b^*$ denotes the ground-truth box coordinates. Intuitively, $\mathcal{T}_x$ indicates the class confidence of a set of predicted boxes surrounding the victim object. Additionally, $\alpha$ is a pre-defined threshold used to quantify the degree of overlap between predictions and the ground-truth box and set to 0.5 by default. In this naive extension, the objective is to consider every predicted box surrounding the victim object as the target of the attack and aim to reverse-engineer a trigger that compromises all of them. We explain this limitation in the following.

**Mis-aligned Loss and ASR.** *Attack Success Rate (ASR)* denotes the percentage of victim samples misclassified as the target label when the inverted trigger is applied (e.g., set at 90%). For *misclassification trigger*, a successful attack is defined as the presence of at least one box predicts the target class in the final detection. *Evasion Attack* is a particular case of misclassification attack where the target label is the background class $\emptyset$. *Cross-entropy (CE)* loss measures the numerical difference between model output (confidence) and the desired prediction. In general, ASR and the CE loss are negatively correlated throughout the optimization process because minimizing the CE loss increases the probability of the target label. If the target label probability surpasses those of other labels, the input is misclassified as the target label. Figure 3(a) illustrates the characteristics of the CE loss and the ASR during trigger inversion for an image classification task.

Such a correlation however may not hold for object detection using a naive trigger inversion Eq. 2. Specifically, Figure 3(b) shows the inversion process on a Faster-RCNN [45] object detection model poisoned by an evasion trigger (TrojAI Round13#91, victim class 47). Note that an evasion trigger

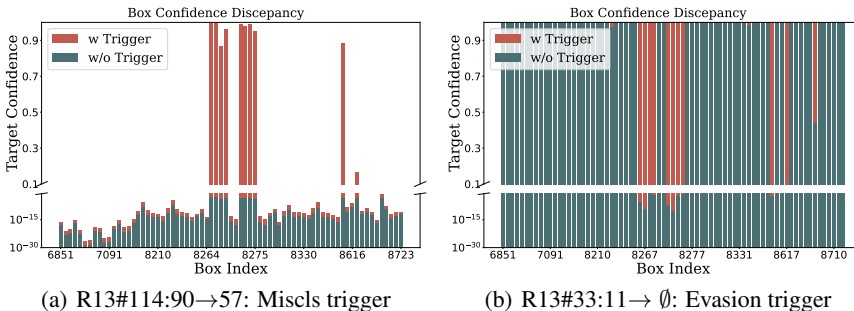

(a) R13#114:90→57: Miscls trigger

(b) R13#33:11→ ∅: Evasion trigger

Figure 4: Target confidence discrepancy w/o the ground-truth trigger

will cause all bounding boxes surrounding the victim object to disappear. Observe that even when the CE loss is small ($\leq 0.1$), the ASR remains low (0%). For a model poisoned by misclassification attack (Round13#12 victim class 3), as illustrated in Figure 3(c), the CE loss remains incredibly high ($\geq 10$) and decreases only slightly during trigger inversion, while the ASR remains low or even decreases. Therefore, the naive trigger inversion fails to find a trigger with high ASR and cannot identify the poisoned model. The observations are similar when extending several state-of-the-art trigger inversion techniques [33, 59, 55, 18] to object detection. For instance, NC [59] only achieves 58% detection accuracy on the TrojAI dataset [1].

**Causes of Misalignment.** We study the misalignment problem by using differential analysis to compare the intermediate outputs of the object detection model with and without the ground-truth trigger, specifically the target label confidence of predicted bounding boxes. Given a sample with a victim object, we collect all predicted boxes with $IoU \geq \alpha$ ($\alpha = 0.5$ means a box is close to the object) around the victim object and track their target label probability change before and after applying the ground-truth trigger. The key observation is that *not every box is equally poisoned*. Figure 4(a) shows the per-box confidence discrepancy for a poisoned model with a misclassification trigger. Although there are a total number of 84 boxes surrounding the victim object, only approximately 12% of them (10 boxes, shown in high red bars) are infected by the ground-truth backdoor trigger. The rest of the boxes only have an average of $\leq 10^{-10}$ probability increase on the target label. For the evasion trigger in Figure 4(b) , there is a slight difference, where 85% of the boxes already have high confidence (on the background class $\emptyset$) even without triggers. The backdoor trigger only elevates the target probability for the remaining 15% of the boxes.

This intuitively explains the misalignment issue of naive trigger reverse-engineering methods. During optimization, the cross-entropy loss of each box is equally aggregated, and the averaged gradient is used to mutate the trigger via back-propagation. However, in the poisoned model, there are only a small fraction of bounding boxes affected by the backdoor trigger. The gradient information from unaffected boxes dominates the optimization direction, causing the inversion to fail. This observation highlights the significance of considering individual bounding boxes separately when devising inversion methods for object detection, which shall *focus on boxes that have a higher likelihood of being compromised but have not been considered yet*.

### 3.2 Our Solution: Trigger Inversion via Gaussian Focus Loss

As discussed in the last section, the goal of our trigger inversion is to discriminate different bounding boxes during optimization. Particularly, we aim to find easy-to-flip boxes that are vulnerable and targeted by injected backdoors. However, it is non-trivial to adaptively select boxes during trigger inversion as the box output is constantly changing.

*Focal Loss* [29] is a widely used metric in object detection for assisting better training. It separates objects into hard-to-classify and easy-to-classify objects based on their training errors, and assigns different weights on their training losses. For the notation simplicity, we use a binary object detection model. Note that it is straightforward to extend it to multi-class cases. Recall $\hat{p}_k$ denotes the output probability of the $k$-th victim box. The CE loss in Eq. 2 for the $k$-th victim box can be simplified as $\mathcal{L}(\hat{p}_k, c_t) = -\log(\hat{p}_k) \cdot c_t = -\log(\hat{p}_k)$. *Focal Loss* is a refined weighted CE loss aiming to solve the imbalanced learning difficulty issue in object detection models. In detail, it assigns larger weights to boxes that perform poorly (*hard examples*), i.e., small $\hat{p}_k$, to enlarge its contribution to the aggregated

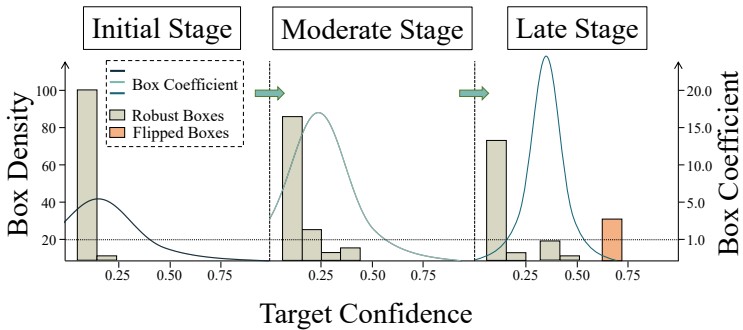

Figure 5: Coefficient dynamics during reverse-engineering a misclassification trigger

loss. It assigns smaller weights to boxes that have already learned well (*easy examples*), i.e., large $\hat{p}_k$:

$$\mathcal{L}_{fl}(\hat{p}_k, c_t) = -\lambda(1 - \hat{p}_k)^\gamma \log(\hat{p}_k), \tag{3}$$

where $\lambda \geq 0, \gamma \geq 1$ ($\lambda = 0.25, \gamma = 2$ suggested in [29]). The coefficient of the $k$-th box will exponentially grow as its probability $\hat{p}_k$ gets close to 0 and vice versa.

Eq. 3 provides insights to address the misalignment problem. Inspired by this, we propose *Gaussian Focus Loss* to calibrate the misaligned loss. As illustrated in Figure 4(a), the backdoor trigger tend to infect only a small portion of victim boxes and leaves remaining largely unchanged. In other words, a trigger will mainly focus on compromising more vulnerable boxes that are easy to flip (*easy example*). Therefore, our goal is essentially opposite to the *Focal Loss* as it aims to pay more attention to the *hard samples*. To avoid the gradient vanishing issue shown in Figure 4(b), a box shall also not be focused as long as it reaches the attack criterion, i.e., target label confidence $\hat{p}_k \geq 0.1$. To summarize, the desired loss shall be able to dynamically capture a set of vulnerable boxes that have not been flipped yet, and assign a large coefficient to encourage the transition. The natural bell shape of the Gaussian Distribution perfectly satisfies our constraints[*]. We formulate our proposed novel *Gaussian Focus Loss* as follows:

$$\mathcal{L}_{gf}(\hat{p}_k, c_t) = -\mathcal{J}(\hat{b}_k, b^*)\phi(\hat{p}_k^\gamma; \hat{\sigma}_k, \hat{\mu}_k) \log(\hat{p}_k)$$
$$where\ \mathcal{J}(\hat{b}_k, b^*) = IoU(\hat{b}_k, b^*),\ \phi(x; \sigma, \mu) = \frac{1}{\sigma\sqrt{2\pi}}e^{-\frac{1}{2}(\frac{x-\mu}{\sigma})^2} \tag{4}$$

It involves two key attributes to measure the vulnerability of a victim box during optimization: the overlapping with the ground-truth object and the confidence towards the target label. Namely, boxes with larger overlapping and higher target label confidence tend to be more vulnerable. The $\hat{\mu}_k$ and $\hat{\sigma}_k$ are the mean and standard deviation of the Gaussian Distribution. Note that we allow two parameters to be co-optimized with the inverted trigger such that we can adjust the range of focused boxes dynamically through the entire trigger inversion procedure. We initialize $\hat{\mu}_k = 0.1$ and $\hat{\sigma}_k = 2$ in this paper. Figure 5 provides a visual explanation of the coefficient dynamics during various optimization steps. Initially, when the victim boxes exhibit low confidence towards the target label, the *Gaussian Focus Loss* assigns slightly larger weights to a considerable number of bounding boxes surrounding the initial value (0.1). As the optimization progresses, the *Gaussian Focus Loss* prioritizes more vulnerable boxes that have higher target label probabilities by decreasing the value of $\sigma$ and increasing the value of $\mu$. In the later stages, the weight coefficients of boxes that have already been flipped are reduced to mitigate the issue of gradient vanishing.

## 3.3 Compromised Label Proposal via Backdoor Leakage

Without knowing the exact victim and target labels, scanning backdoors is time-consuming. To reduce the time cost, pre-processing is proposed [49, 54] to quickly select a small set of promising compromised label pairs based on the widely existing *backdoor leakage* phenomena: the poisoned

---

[*]Any distributions characterized by centralized peaks are suitable for our intended purpose. We explore alternatives in Appendix D.

Table 1: Comparison to trigger inversion based methods

| D | M | Metric | ABS | NC | NC* | Pixel | Pixel* | Tabor | Tabor* | **DJANGO** |
|---|---|---|---|---|---|---|---|---|---|---|
| Traffic Sign Synthesis | SSD | Precision | 0.7143 | 0.6250 | 0.6522 | 0.4400 | 0.6666 | 0.3300 | 0.4000 | 0.8000 |
| | | Recall | 0.6250 | 0.9375 | 0.9375 | 0.5000 | 0.2500 | 0.3750 | 0.6250 | 1.0000 |
| | | ROC-AUC | 0.7109 | 0.6250 | 0.6641 | 0.6250 | 0.7083 | 0.5400 | 0.6725 | **0.9160** |
| | | Overhead(s) | 409.23 | >3600 | 1180.52 | >3600 | 953.42 | >3600 | 902.10 | 861.30 |
| | F-RCNN | Precision | 0.6667 | 0.3478 | 0.6364 | 0.5512 | 0.6250 | 0.6621 | 0.7125 | 0.8571 |
| | | Recall | 0.3750 | 1.0000 | 0.8750 | 0.6110 | 0.6250 | 0.6012 | 0.6250 | 0.7500 |
| | | ROC-AUC | 0.5352 | 0.6094 | 0.6953 | 0.6721 | 0.7500 | 0.6425 | 0.6820 | **0.8750** |
| | | Overhead(s) | 753.89 | >3600 | 2799.42 | >3600 | 2466.08 | >3600 | 2562.59 | 2128.99 |
| | DETR | Precision | - | 0.3478 | 0.3478 | 0.4000 | 0.5000 | 0.2307 | 0.5000 | 0.8750 |
| | | Recall | - | 1.0000 | 1.0000 | 0.5000 | 0.3750 | 0.3750 | 0.6250 | 0.8750 |
| | | ROC-AUC | - | 0.5312 | 0.5312 | 0.5000 | 0.6660 | 0.2500 | 0.7600 | **0.9160** |
| | | Overhead(s) | - | >3600 | 691.27 | >3600 | 294.32 | >3600 | 275.50 | 228.92 |
| DOTA | SSD | Precision | 0.6000 | 0.3636 | 1.0000 | 0.3300 | 0.3000 | 0.5000 | 0.3333 | 1.0000 |
| | | Recall | 0.7500 | 1.0000 | 0.1250 | 0.2500 | 0.7500 | 0.2500 | 0.5000 | 1.0000 |
| | | ROC-AUC | 0.5000 | 0.5312 | 0.6875 | 0.3750 | 0.3333 | 0.5000 | 0.6250 | **1.0000** |
| | | Overhead(s) | 424.16 | >3600 | 675.32 | >3600 | 529.11 | >3600 | 618.72 | 678.48 |
| | F-RCNN | Precision | 0.8000 | 0.3636 | 0.3636 | 0.3330 | 0.5000 | 0.3300 | 0.3300 | 0.8000 |
| | | Recall | 1.0000 | 1.0000 | 1.0000 | 0.5000 | 0.7500 | 0.5000 | 0.5000 | 1.0000 |
| | | ROC-AUC | 0.8750 | 0.5625 | 0.5625 | 0.5000 | 0.6666 | 0.5000 | 0.5000 | **0.9160** |
| | | Overhead(s) | 1127.01 | >3600 | 2866.14 | >3600 | 1565.82 | >3600 | 1602.10 | 1425.25 |
| COCO | SSD | Precision | 0.5167 | 0.5231 | 0.8333 | 0.7500 | 1.0000 | 0.6120 | 0.7500 | 0.9696 |
| | | Recall | 0.9688 | 1.0000 | 0.1471 | 0.2500 | 0.2500 | 0.5620 | 0.5300 | 0.8888 |
| | | ROC-AUC | 0.5584 | 0.5404 | 0.5579 | 0.6200 | 0.7000 | 0.6419 | 0.6820 | **0.9305** |
| | | Overhead(s) | 273.23 | >3600 | 2119.34 | >3600 | 1788.50 | >3600 | 1688.19 | 1476.01 |

model's behavior on victim samples tends to shift towards the target label even without the appearance of the backdoor trigger. Therefore, we can feed clean samples from each class and pick the most likely target label by observing the model's output distribution shift. Specifically, given a set of samples $\{x_i\}_{i=1}^n$ from class $c_i$, we collect top-$h$ predicted classes of each predicted box for each sample $x_i$. Denote $\omega_j$ as the frequency of class $c_j$ appearing in the total box predictions. We consider a pair $\{c_i, c_j\}$ as a compromised label pair if $\omega_j \geq \omega$. In this paper, we make a trade-off by setting $h = 5$ and $\omega = 0.5$. We also evaluate the sensitivity of hyper-parameters in Section 4.2.

# 4  Evaluation

All the experiments are conducted on a server equipped with two Intel Xeon Silver 4214 2.40GHz 12-core processors, 192 GB of RAM, and a NVIDIA RTX A6000 GPU.

**Models and Datasets.** We conduct the experiments on models from TrojAI [1] round 10 and round 13, which consists of a total of 168 models. Approximately half of these models are clean, while the other half are attacked. Our evaluation covers 3 existing object detection image datasets, including COCO [30], Synthesized Traffic Signs [1], and DOTA_v2 [11] on three representative object detection architectures: single-stage detector: SSD [32], two-stage detector: Faster-RCNN [45] and vision transformer based detector: DETR [2]. Please refer to Appendix A for more detailed description.

**Attack Settings.** We conducted an evaluation of DJANGO by subjecting it to two types of backdoor attacks commonly observed in object detection models: Misclassification and Evasion attacks [3, 38] Misclassification attack means to flip the prediction of objects from the victim class to the target class, when the backdoor trigger is stamped on the input image. On the other hand, evasion triggers tend to suppress the recognition of victim objects, such that the model will consider them as the background. It is noteworthy that the evasion trigger can be regarded as a particular instance of the misclassification trigger, where the target label is set as the background class $\emptyset$. As a result, its detection can be performed uniformly. Besides, we leverage stealthy polygons with different colors and textures as backdoor triggers, which are widely evaluated in many backdoor detection papers [59, 17, 49].

**Evaluation Metrics.** In our evaluation of backdoor detection methods, we employ four well-established metrics: Precision, Recall, ROC-AUC, and Average Scanning Overheads for each model. The unit of measurement we use is seconds (s), and we set a maximum threshold of 1 hour (3600 s) for all the methods being evaluated. If the scanning process exceeds 3600 seconds, it is terminated, and we rely on the existing results for making predictions. Appendix B present more details.

**Baseline Methods** We compare DJANGO against 6 baseline methods, including 4 trigger inversion based methods (i.e., NC [59], ABS [33], Pixel [55] and Tabor [18]), and 2 meta-classification based methods (i.e., MNTD [69] and MF [21]). For meta classification based methods that involve training, we have performed 5-fold cross-validation and reported the validation results exclusively. For the 4 inversion-based scanners, we adopt their objective functions as described in Eq. 2 and keep their other designs unchanged. To ensure a fair comparison, we use the same setting for common optimization hyper-parameters for all inversion based methods. We determine the optimal threshold for the size of inverted triggers as the detection rule for[59, 55, 18]. For ABS, we follow its original technique and use REASR as the decision score.

## 4.1 Detection Performance

**Comparison to Trigger Inversion based Methods.** Table 1 shows the results for trigger inversion based methods, where the first two columns denote the datasets and model architectures, the third column the evaluation metrics, and the subsequent columns the baselines. Additionally, considering that NC [59], Pixel [55], and Tabor [18] need to scan all possible victimtarget pairs, which can exceed the time limit, we have employed our warm-up pre-processing technique (Section 3.3). We compare the enhanced versions of these methods (NC*, Pixel*, Tabor*) with DJANGO. Note that we have not evaluated ABS on DETR models, as the original version is specifically designed for CNN-based models and not transformers.

DJANGO consistently outperforms all the baselines, with the best results are highlighted in bold. The average ROC-AUC of DJANGO is 0.913, while the highest ROC-AUC achieved by

Table 2: Comparison to meta classifiers

| D | M | Metric | MNTD | MF | DJANGO |
|---|---|--------|------|-----|--------|
| Traffic Sign Synthesis | SSD | Precision | 0.4545 | 0.4000 | 0.8000 |
| | | Recall | 0.3125 | 0.5000 | 1.0000 |
| | | ROC-AUC | 0.3750 | 0.6563 | **0.9160** |
| | F-RCNN | Precision | 0.5652 | 0.5000 | 0.8571 |
| | | Recall | 0.8125 | 0.1250 | 0.7500 |
| | | ROC-AUC | 0.7695 | 0.6797 | **0.8750** |
| | DETR | Precision | 0.1250 | 0.1250 | 0.8750 |
| | | Recall | 0.1250 | 0.1250 | 0.8750 |
| | | ROC-AUC | 0.1719 | 0.2890 | **0.9160** |
| DOTA | SSD | Precision | 0.3333 | 0.3333 | 1.0000 |
| | | Recall | 0.1250 | 0.5000 | 1.0000 |
| | | ROC-AUC | 0.3906 | 0.4375 | **1.0000** |
| | F-RCNN | Precision | 1.0000 | 0.3333 | 0.8000 |
| | | Recall | 0.5000 | 0.5000 | 1.0000 |
| | | ROC-AUC | 0.6562 | 0.6875 | **0.9160** |
| COCO | SSD | Precision | 0.8182 | 0.7000 | 0.9696 |
| | | Recall | 0.5625 | 1.0000 | 0.8888 |
| | | ROC-AUC | 0.8144 | 0.8163 | **0.9305** |

the baselines is 0.875, which is the result obtained by ABS on DOTA and F-RCNN. The average ROC-AUC of these baselines is nearly 0.592. Furthermore, our warm-up pre-processing technique has proven effective in enhancing the scanning performance of the baselines, resulting in an improvement of 0.05 to 0.15 in terms of ROC-AUC and a significant reduction in overheads from over 3600 seconds to a range of 275 to 2800 seconds. Despite these advancements, the enhanced versions of the baselines are still unable to surpass the performance of DJANGO, with ROC-AUC gaps ranging from 0.15 to 0.50.

Additionally, we have observed that the overheads incurred by DJANGO are consistently lower than those of NC, Pixel, and Tabor, even when these methods are equipped with our warm-up preprocessing technique. However, ABS generally achieves lower overheads than DJANGO, primarily because ABS only performs trigger inversion for the top-10 target classes according to the original configuration. Nonetheless, it is important to note that DJANGO significantly outperforms ABS in terms of ROC-AUC, which is the more critical metric for our evaluation. We further illustrate the inverted trigger by DJANGO in Figure 6. It is evident that the inverted trigger produced by DJANGO closely resembles the ground-truth injected trigger in terms of both shape and color. The overall success of DJANGO can be attributed to our well-designed warm-up pre-processing and the novel *Gaussian Focus Loss* 4. These techniques play a crucial role in achieving the superior performance and efficiency demonstrated by DJANGO compared to the baselines.

**Comparison to Meta Classifiers.** Table 2 provides a comparison between meta classifiers and DJANGO. Since meta classifiers require only a few seconds to scan a model, we have omitted the overheads from the table. Observe that baselines achieve decent results in the COCO dataset with over 0.81 ROC-AUC, while only make nearly 0.6 ROC-AUC on other datasets. This discrepancy can be attributed to the fact that there are over 60 models available in the COCO dataset, whereas there are only 30 models in the other datasets. Consequently, the performance of the meta classifiers heavily relies on the availability of a large number of training models. Unfortunately, obtaining such a

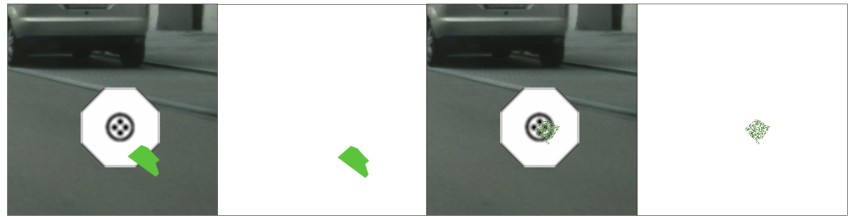

Figure 6: Visual similarity between GT and DJANGO inverted triggers.

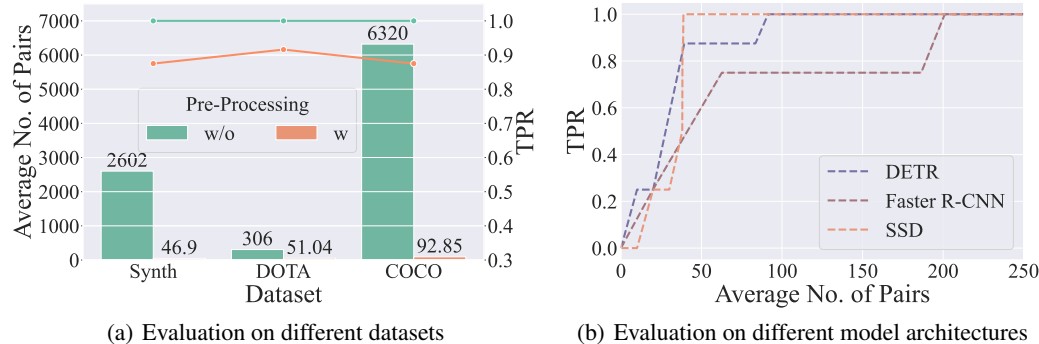

(a) Evaluation on different datasets

(b) Evaluation on different model architectures

Figure 7: Evaluation of label proposal pre-processing

significant number of models in real-world scenarios is a challenging task. In contrast, DJANGO does not require access to training models. Moreover, it achieves a ROC-AUC of over 0.91, outperforming the meta classifiers even when they are trained on a sufficient number of models in the COCO dataset. This highlights the effectiveness and robustness of DJANGO in detecting backdoor attacks, surpassing the performance of meta classifiers.

**Evaluation on Advanced Object Detection Backdoor Attacks.** We also evaluate DJANGO on two advanced object detection backdoor attacks [3, 5, 28]. DJANGO achieves 0.8 and 0.9 ROC-AUC on detecting these attacks. Appendix E presents more discussion.

## 4.2 Evaluation of Label Proposal Pre-processing

In this section, we evaluate the effectiveness of our label proposal pre-processing introduced in Section 3.3. Results are presented in Figure 7. As shown in Figure 7(a), where the x-axis denotes the dataset, the left y-axis denotes the average number of pairs and the right one the TPR (True Positive Rate) of the selected pairs. In the case of an attacked model, if the selected pairs after pre-processing contain the ground-truth victim-target pair, it is considered a true positive. TPR is calculated as the ratio of true positives to all positives. The green bars represent the average number of pairs without pre-processing, while the red bars represent the number of pairs after pre-processing. We observe that across all datasets, our pre-processing technique significantly reduces the number of scanning pairs by 83% to 98%. The curves represent the TPR, showing that our pre-processing results in TPRs of over 88% across different datasets. These results indicate that our pre-processing technique effectively reduces the number of pairs to a remarkably low level while maintaining a high detection rate. The experiment is conducted by selecting various values of $h$ ranging from 1 to 10 (representing the top $h$ class labels) and $\omega$ ranging from 0.1 to 0.8 (representing the frequency threshold). We record the average number of pairs and true positive rate (TPR) under this configuration. The results are depicted in Figure 7(b). we observe that SSD and DETR models require fewer than 100 pairs to reach high TPR, while Faster-RCNN models require around 200 pairs. One potential reason for this difference is that Faster-RCNN requires a two-stage training process where the backdoor signal is less prominent compared to single-stage training methods in SSD and DETR.

Table 3: Ablation study

| Method | SSD | | Faster-RCNN | | DETR | |
|---|---|---|---|---|---|---|
| | ROC-AUC | Overhead(s) | ROC-AUC | Overhead(s) | ROC-AUC | Overhead(s) |
| DJANGO | **1.0000** | **859.59** | **0.8750** | **2120.10** | **0.8750** | 227.55 |
| DJANGO - GF Loss | 0.7500 | 860.24 | 0.7500 | 2135.59 | 0.7500 | **226.88** |
| DJANGO - Pre-Processing | 0.6250 | >3600 | 0.5000 | >3600 | 0.6250 | >3600 |

## 4.3 Ablation Study

Our ablation experiments are conducted on the synthesized traffic sign dataset with 3 different model architectures. We include 8 models for each architecture, with half clean and half attacked.

There are two key designs of DJANGO to ensure the detection effectiveness, i.e., GF Loss and Pre-Processing. We conduct ablation study to assess the contribution of these components respectively. Results are shown in Table 3, where each row corresponds to a specific ablation method. The results clearly demonstrate the critical importance of both GF Loss and Pre-Processing in DJANGO. Without these components, the detection performance of DJANGO significantly degrades by more than 0.13 ROC-AUC. Pre-Processing also plays a crucial role in reducing the detection overhead. Please refer to sec D for hyper-parameter sensitivity analysis.

## 4.4 Adaptive Attack

In this section, we evaluate the performance of DJANGO in the context of adaptive attacks. The effectiveness of DJANGO relies on the confidence levels assigned to the potential boxes, as these determine the coefficients used for trigger inversion. Consequently, an adaptive attack strategy could

Table 4: Adaptive attack

| Attack Conf. | Clean mAP | ASR_0.1 | ASR_0.6 | ASR_0.8 | DJANGO |
|---|---|---|---|---|---|
| No Attack | 0.8766 | 0.0047 | 0.0023 | 0.0000 | 0.0 |
| 0.80 | 0.8700 | 1.0000 | 1.0000 | 0.9812 | 1.0 |
| 0.60 | 0.8708 | 1.0000 | 1.0000 | 0.4014 | 1.0 |
| 0.51 | 0.8685 | 1.0000 | 0.5728 | 0.0446 | 0.6 |

involve reducing the confidence of objects containing the trigger, thereby potentially causing DJANGO to assign lower coefficients to these poisoned boxes. To evaluate the adaptive attack scenario, we conduct experiments using a 5-class synthesized traffic sign dataset with the SSD model. We train several models with misclassification attacks, where the victim class is to 0 and the target class 4. In order to simulate low confidence attacks, we reduced the ground-truth confidence of the target class to values of 0.8, 0.6, and 0.51, while setting the confidence of the victim class to 0.2, 0.4, and 0.49, respectively. After sufficient training, we applied DJANGO to these models. The results are presented in Table 4, where we recorded the clean mAP and the ASRs at different score thresholds. For instance, ASR_0.1 indicates the ASR when the prediction confidence is higher than 0.1. From the results, we can see that low confidence attacks lead to lower ASRs, especially when the score threshold for measuring the ASR is set high. Furthermore, DJANGO is successful in detecting attacked models with target confidences of 0.8 and 0.6, but fails when the target confidence is 0.51 (threshold set at 0.8). These results suggest that DJANGO is effective against most low-confidence attacks, but its performance may degrade to some extent when the target confidence is very low.

## 5 Limitation

Our primary focus in this paper is on attacks that use static polygon triggers, which are more feasible in real-world scenarios. How to effectively inject more complex triggers [42, 9, 48] in object detection models is still an open question. We leave it to future work.

## 6 Conclusion

In this paper, we present DJANGO, the first trigger inversion framework on detecting backdoors in object detection models. It is based on a novel *Gaussian Focus Loss* to tackle the loss misalignment issue in object detection trigger inversion. Extensive experiments demonstrate the effectiveness and efficiency of DJANGO compared to existing baselines.

# 7 Acknowledgment

We thank the anonymous reviewers for their constructive comments. We are grateful to the Center for AI Safety for providing computational resources. This research was supported, in part by IARPA TrojAI W911NF-19-S-0012, NSF 1901242 and 1910300, ONR N000141712045, N000141410468 and N000141712947. Any opinions, findings, and conclusions in this paper are those of the authors only and do not necessarily reflect the views of our sponsors.

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

# Appendix

## A  Details of datasets and architectures

### A.1  Object Detection Image Dataset

**COCO (Common Objects in Context) [30]** dataset is widely used for object detection tasks. It contains 80 object categories, including people, animals, vehicles and more. Each image can contain multiple instances of objects, providing ample opportunities for training and evaluating models capable of detecting and segmenting objects in complex scenes.

**Synthesized Traffic Sign** dataset is designed by TrojAI [1] which focuses on traffic sign detection, featuring various types of traffic signs commonly encountered in real-world scenarios. There are in total over 4000 different traffic signs. Each model is trained on a randomly sampled subset of classes. The number of classes within these subsets exhibits variability, ranging from as few as 2 to a maximum of 128.

**DOTA (Detection in Aerial Images)** dataset is designed for object detection in aerial images which consists of high-resolution images captured by aerial platforms. It contains 18 categories, including plane, ship, storage tank, baseball diamond and more. Its large-scale, fine-grained annotations, and challenging scenarios make it an ideal benchmark for evaluating and developing algorithms capable of detecting objects in aerial images accurately.

### A.2  Architecture

We evaluate our method on three well-known model architectures:, i.e., SSD [32], Faster-RCNN [45], and DETR [2]. SSD (Single Shot MultiBox Detector) [32] is a popular object detection model which utilizes a series of convolutional layers to detect objects at multiple scales and aspect ratios. Faster-RCNN [45] is another widely adopted object detection model that combines region proposal generation with a region-based CNN for object detection. DETR (DEtection TRansformer) [2] is a state-of-the-art object detection model that utilizes a transformer-based architecture. It replaces the conventional two-stage approach with a single-stage end-to-end detection framework.

### A.3  Model Dataset

**TrojAI** [1] initiative, spearheaded by IARPA, encompasses a multi-year, multi-round program. Its overarching objective revolves around the development of scalable and dependable automatic backdoor detection tools, specifically targeting the identification of backdoor trojans within Deep Learning models across diverse modalities. Presently, the program consists of a total of 13 rounds, each with distinct focuses and tasks. The initial four rounds and the eleventh round center their efforts on detecting trojans present in image classification models. In contrast, rounds five through nine concentrate on transformer models employed in various NLP tasks, including Sentiment Analysis, Named Entity Recognition, and Question Answering. Round twelve dedicates itself to the detection of backdoors in neural network-based PDF malware detection. Finally, rounds ten and thirteen direct their attention towards object detection models. For the evaluation of models, we exclusively utilize the training sets from rounds 10 and 13. As shown in Table 5, our evaluation entails 72 models trained on the Synthesis Traffic Sign dataset, encompassing all three model architectures. Among these models, 48 are benign, while 24 are deliberately poisoned, with an equal distribution of triggers for misclassification and evasion. Concerning the DOTA models, there exist two architectures, namely SSD and Faster-RCNN, resulting in a total of 24 models, including 16 benign models and 4 each poisoned with misclassification and evasion triggers. All COCO models adopt the SSD architecture, with a distribution of 36 clean models and 18 models poisoned by both misclassification and evasion triggers. To provide further elucidation,it is important to note that the poison rate varies within the range of 0.1% to 8% across diverse models sourced from TrojAI r10 and r13. Similarly, the trigger size exhibits a range of 1x1 to 22x22, representing a scale of 0.001% to 0.7% relative to the input dimensions. Pertaining to the hyper-parameters utilized in model training, the learning rate is stochastically assigned, spanning from $1.56e-8$ to $1e-4$ across different models. The number of epochs for training spans from 6 to 100, while the batch size ranges from 4 to 32. As for the model performance metrics, the average clean mAP across the models attains a value of 0.7979, while the average poison mAP stands at 0.7680. The trigger's polygonal structure is characterized by varying

edge counts, ranging from 3 to 8. Furthermore, each individual trigger is endowed with randomly generated color and texture attributes. Concrete settings can be found Round13[*] and Round10[*].

## B    Details of evaluation metrics

In our evaluation of backdoor detection methods, we employ four well-established metrics: Precision, Recall, ROC-AUC, and Average Scanning Overheads for each model. Precision quantifies the accuracy of a detection method by measuring the proportion of correctly identified positive instances among all predicted positives. In our case, we consider attacked models as positive instances and benign models as negatives. A higher precision indicates a lower rate of falsely identifying benign models as attacked. Recall, on the other hand, assesses the effectiveness of the detection method in correctly identifying positive instances. It measures the proportion of true positives among all actual positives. A higher recall suggests that the detection method is capable of identifying a significant portion of attacked models. ROC-AUC (Receiver Operating Characteristic - Area Under the Curve) plots the true positive rate against the false positive rate at various threshold values and calculates the area under the curve. A value of 1 indicates perfect classification, while a value of 0.5 indicates that the method is no better than random guessing. We also consider the overhead of the detection method, which quantifies the average time required to scan a single model. We use seconds (s) as the unit of measurement and set a maximum threshold of 1 hour (3600 s). If the scanning process exceeds 3600 seconds, it is terminated, and we rely on the existing results for making predictions. Low overhead signifies high efficiency of the method. By employing these four metrics, we aim to comprehensively evaluate the performance and efficiency of the backdoor detection methods. It is worth noting that the time limit we have set for scanning models is deliberately conservative when compared to the thresholds established in different rounds of the TrojAI competition. For example, in round 13, participants are granted a generous 30-minute duration for scanning a single model. To surpass the official benchmarks set in each round, a more aggressive and precise pre-processing approach may be necessary.

## C    Details of Baseline Methods

In this section, we introduce more details of baseline methods, including NC [59], Tabor [18], ABS [33], Pixel [55], Matrix Factorization(MF) [21] and MNTD [69].

**NC** [59] adopts a specific trigger inversion approach for each class and considers a model to be attacked if it is able to generate an effective yet extremely small trigger for a target class. **Tabor** [18] enhances NC by incorporating additional well-designed regularization terms, such as penalties for scattered triggers, overlaying triggers, and blocking triggers. These additions aim to improve the reconstruction of injected triggers. **Pixel** [55] introduces a novel inversion function that generates a pair of positive and negative trigger patterns. This approach achieves better detection performance compared to NC. **ABS** [33] employs a stimulation analysis to identify compromised neurons, which serves as guidance for trigger inversion. ABS considers a model to be attacked if it can invert a trigger that achieves a high reconstructed ASR (REASR).

To the best of our knowledge, there is no existing detection methods for object detection models. Therefore, we perform straight-forward but reasonable adaption to these existing methods designed on image classification tasks, such that they are able to work against backdoor attacks on object detection models. Specifically, the original objective of NC, Tabor, and Pixel is to invert small triggers while maintaining their effectiveness (high ASR). In our adaptation, we retain their design principles but re-define the ASR to align with object detection models, as explained in Section 3.1. Additionally, we introduce a threshold for the size of inverted triggers, enabling the differentiation between benign and attacked models. For ABS, we adhere to its original technique but employ the re-defined ASR as the optimization goal, and use REASR as the decision score. By employing these adaptations, we aim to enhance the detection capabilities of these existing methods specifically for backdoor attacks on object detection models.

No modifications or adaptations are needed for meta classification-based methods when applied to object detection models. MNTD [69] trains a set of queries and a classifier to discern the feature-space

---

[*]https://pages.nist.gov/trojai/docs/object-detection-feb2023.html#object-detection-feb2023
[*]https://pages.nist.gov/trojai/docs/object-detection-jul2022.html#object-detection-jul2022

Table 5: Dataset details

| Image Dataset | Model Source | | Architecture | | | Number of Models | | |
|---|---|---|---|---|---|---|---|---|
| | Round10 | Round13 | SSD | Faster-RCNN | DETR | Benign | Miscls Attack | Evasion Attack |
| Synthesis Traffic Sign | ✗ | ✓ | ✓ | ✓ | ✓ | 48 | 12 | 12 |
| DOTA | ✗ | ✓ | ✓ | ✓ | ✗ | 16 | 4 | 4 |
| COCO | ✓ | ✗ | ✓ | ✗ | ✗ | 36 | 18 | 18 |

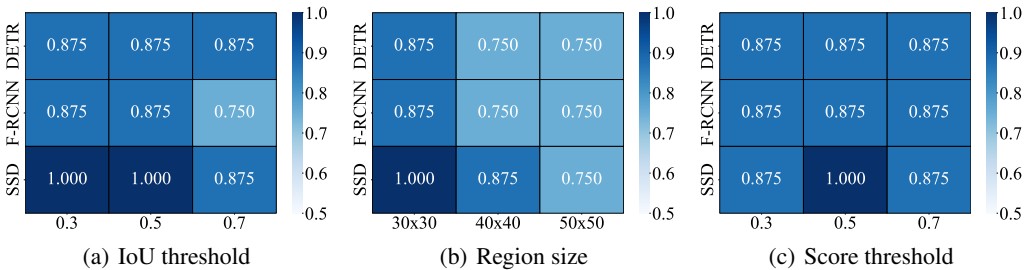

(a) IoU threshold          (b) Region size          (c) Score threshold

Figure 8: Hyper-parameter sensitivity.

distinctions between clean and attacked models. MF [21] directly trains a classifier on model weight features using specialized feature extraction techniques, i.e., matrix factorization. These methods solely rely on the feature extraction networks commonly utilized in both image classification and object detection models. As a result, MNTD and MF can be directly employed to detect backdoor attacks in object detection models without the need for additional adjustments or modifications.

We collect the Precision, Recall, ROC-AUC and Overheads for each method across various datasets and model architectures. To ensure a fair comparison, we have conducted a search to determine the optimal thresholds for different decision scores associated with each method (trigger size for NC, Tabor, Pixel, REASR for ABS and output confidence for meta-classifiers). These thresholds are chosen to maximize accuracy. Besides, we set a fixed number of optimization steps for scanning a pair of victim-target label (100) for all inversion based baselines. For meta classification based methods that involve training, we have performed 5-fold cross-validation and reported the validation results exclusively.

# D Hyper-parameter Sensitivity Analysis

To assess the sensitivities of the hyper-parameters used in DJANGO, we conduct experiments as described in Section 4.3.

**IoU Thresholds.** We evaluate the IoU threshold used to calculate the ASR of inverted triggers. The results are summarized in Figure 8(a), where each row corresponds to a different model architecture, and each column represents a different choice of IoU threshold. It can be observed that IoU thresholds of 0.3 and 0.5 generally yield good performance. However, a threshold of 0.7 tends to degrade the performance, possibly due to the inverted triggers interfering with the bounding box predictions.

**Region Size.** The impact of different regional initialization sizes is evaluated and the results are presented in Figure 8(b). Among the various choices, a region size of 30×30 consistently achieved the best performance. This is because larger initialization sizes tend to result in more false positive cases.

**Score Threshold.** Different score thresholds are tested when computing the ASR of inverted triggers. The results, shown in Figure 8(c), indicate that a score threshold of 0.5 generally leads to the best performance across all model architectures. This choice represents a trade-off between false positives and false negatives. Higher score thresholds may introduce more false negatives, as the inverted trigger may not have high confidence similar to the injected one. On the other hand, lower score thresholds may result in more false positives. Thus, a moderate value of 0.5 provides the optimal balance.

Table 6: Hyper-parameter sensitivity for Gaussian Focus Loss initial values

| $(\hat{\mu}_k, \hat{\sigma}_k)$ | ROC-AUC | | |
|---|---|---|---|
| | SSD | Faster-RCNN | DETR |
| **(0.1, 2)** | 1.0000 | 1.0000 | 0.8750 |
| (0.1, 1) | 1.0000 | 1.0000 | 0.8750 |
| (0.1, 3) | 1.0000 | 0.8750 | 0.8750 |
| (0.2, 2) | 0.8750 | 0.7500 | 0.8750 |
| (0.2, 1) | 0.8750 | 0.8750 | 0.7500 |
| (0.2, 3) | 0.8750 | 0.8750 | 0.8750 |

Table 7: Effectiveness on different weight scheme distributions

| Method | ROC-AUC |
|---|---|
| **DJANGO** + GF Loss | 0.9500 |
| **DJANGO** + LP Loss | 0.9000 |
| Tabor* | 0.6500 |
| NC* | 0.6000 |
| Pixel* | 0.7000 |

Table 8: Effectiveness on complex attacks

| Attack type | ROC-AUC |
|---|---|
| Object-appearing Attack | 0.8000 |
| Composite Attack | 0.9000 |

These experiments allowed us to gain insights into the sensitivities of the hyper-parameters in DJANGO, enabling us to make informed choices for achieving optimal performance.

**GF Loss Initialization.** We evaluate the impact of two hyper-parameters (initial mean $\hat{\mu}_k$ and variance $\hat{\sigma}_k$) in the Eq. 4 on DJANGO. We randomly sample 8 models (4 trojan and 4 benign) for each architecture that was trained on the synthesized traffic sign dataset. Besides the default values we report in the paper ($\hat{\mu}_k = 0.1$, $\hat{\sigma}_k = 2$), we set 5 more groups of initial values and report the detection performance. As shown in Table 6, DJANGO remains effective under different initialization values.

**Weight Scheme Distribution Choice.** We attempted to substitute the Gaussian distribution with the Laplace distribution, another commonly employed probability distribution. Our experimentation involved 20 models trained on the traffic sign synthetic dataset, comprising 10 clean models and 10 models poisoned with misclassification triggers. The outcomes of these experiments are presented in Table 7. It is evident that with the Laplace focus loss, DJANGO maintains a high level of effectiveness, achieving an ROC-AUC of 0.9, whereas baseline methods only achieve 0.7. We hypothesize that the 0.05 performance decline, in comparison to the Gaussian Focus Loss, may be attributed to the sharper peak of the Laplace distribution when contrasted with the Gaussian distribution. Within our context, a reduced number of bounding boxes exhibiting moderate confidences are allocated larger coefficients in different stages, thereby potentially diminishing the emphasis on compromised boxes.

## E Evaluation on Complex Object Detection Backdoor Attacks

We further evaluate DJANGO on two advanced object detection backdoor attacks: Object-appearing and Composite attacks [5, 3, 28]. Object-appearing triggers aim to yield target class object bounding boxes on the background. According to our definition, the object-appearing attack is a special case of the misclassification attack with the background $\emptyset$ class as the victim class. To demonstrate the effectiveness of DJANGO on object-appearing attacks, we evaluate DJANGO on 10 Baddet models and 10 Clean-image backdoor models during rebuttal. For each type of attack, we mix 5 clean models with 5 poisoned models with object-appearing triggers. The evaluation results are shown in Table 8. DJANGO achieves 0.8 ROC-AUC on both Baddet and clean-image object appearing attacks.

For the composite attack, we conducted an evaluation of DJANGO using 10 models that were trained on the traffic sign synthesis dataset. This set included 5 clean models and 5 trojan models poisoned by composite attack. As indicated in Table 8, DJANGO achieved an 0.9 ROC-AUC for detecting composite attacks. It's worth noting that the composite attack does not rely on an explicit trigger.

Instead, it leverages a clean object A to serve as the trigger for attacking another object B. Interestingly, DJANGO is capable of effectively reversing this process, essentially identifying a trigger that closely mimics the pattern of object A. This ability enables DJANGO to detect composite backdoors with a high level of accuracy.

