# OpenReview forum: "Django: Detecting Trojans in Object Detection Models via Gaussian Focus Calibration"
_NeurIPS.cc/2023/Conference — NeurIPS 2023 poster_

### Official Review · Reviewer_tRsV · 2023-07-03

**Soundness:** 2 fair
**Presentation:** 2 fair
**Contribution:** 2 fair
**Rating:** 5
**Confidence:** 3

**Summary:**

The paper proposes the first object detection backdoor detection framework Django (Detecting Trojans in Object Detection Models via Gaussian Focus Calibration). It leverages a dynamic Gaussian weighting scheme that prioritizes more vulnerable victim boxes and assigns appropriate coefficients to calibrate the optimization objective during trigger inversion.

The experimental results show the superiority of Django over some state-of-the-art baselines.

**Strengths:**

The authors found that the poison effect can vary significantly across bounding boxes in object detection models due to its dense prediction nature, leading to an undesired optimization objective misalignment issue for existing trigger reverse-engineering methods.

The authors propose a trigger inversion-based backdoor detection framework for object detection: DJANGO (Detecting Trojans in Object Detection Models via Gaussian Focus Calibration). It features a Gaussian Focus Loss to calibrate the misaligned loss during inversion by dynamically assigning weights to individual boxes based on their vulnerability.

The authors claim that equipped with a label proposal pre-processor, DJANGO is able to quickly identify malicious victim-target labels and effectively invert the injected trigger lies in the backdoored model.

The experimental results show Django outperforms some state-of-the-art baselines on four metrics: Precision, Recall, ROC-AUC, and Average Scanning Overheads for each model.

**Weaknesses:**

Can the authors explain why the authors claim that Django is the first trigger inversion-based backdoor detection framework for object detection while there have been also many popular and effective trigger inversion backdoor scanning architectures/methods such as [51, 49, 15, 31]? Do you mean that the proposed framework is the first one for object detection? How about [3] mentioned in the paper?

Inspired by the Focal loss [25], the authors propose Gaussian Focus Loss. However, It is unclear (i) how the proposed Gaussian Focus Loss work to solve object detection backdoor detection, and (ii) how the parts in this Gaussian Focus Loss interact to dynamically capture a set of vulnerable boxes that have not been flipped yet, and assign a large coefficient to encourage the transition.

The setting of the training process of the proposed framework and baselines are not mentioned clearly.

**Questions:**

Please see the questions in the weaknesses section.

Some further questions for improvements:

What is the proposed trigger inversion-based backdoor detection framework? Can the authors show a visualization of a flow or a pipeline or an algorithm of this framework including the main components?

Can the authors explain the setting of the training process (e.g., the models configuration and the values of any hyper-parameters used) of the proposed method and baselines?

**Limitations:**

The limitations of the work are not mentioned in the paper.

---

> ### Author Rebuttal · Authors · 2023-08-10
>
> **[W1] Clarification of the Django**
>
> We are sorry for the confusion. [51, 49, 15, 31] are initially designed for detecting backdoor on image classification models. According to our evaluation(main text, line 268-line 291, Table 1), existing trigger inversion techniques all have limited performance when adapted on object detection models due to the misalignment issue presented in main text Sec 3.1.  In Baddet[3], the proposed Detection Cleanse is a backdoor sample detection technique, which has a completely different threat model and goal with model-level backdoor detection. Please refer to a more detailed discussion in global response A1.  To the best of our knowledge, the proposed Django is the first model-level trigger inversion-based backdoor detection framework tailored for deep learning object detection models. We will further clarify in the revision.
>
> ---
>
> **[W2] Intuition of Gaussian Focus Loss**
>
> Please refer to **global response A2**.
>
> ---
>
> **[W3&Q2] Model & Baseline configuration details**
>
> In Appendix A, we provide comprehensive details regarding the datasets and model architectures employed in this study. To provide further elucidation, it is important to note that the poison rate varies within the range of 0.1% to 8% across diverse models sourced from TrojAI r10 and r13. Similarly, the trigger size exhibits a range of 1x1 to 22x22, representing a scale of 0.001% to 0.7% relative to the input dimensions. Pertaining to the hyper-parameters utilized in model training, the learning rate is stochastically assigned, spanning from 1.56e-08 to 1e-04 across different models. The number of epochs for training spans from 6 to 100, while the batch size ranges from 4 to 32. As for the model performance metrics, the average clean mAP across the models attains a value of 0.7979, while the average poison mAP stands at 0.7680. We assess TrojAI models using polygon triggers, encompassing two distinct attack types: misclassification and evasion attacks. The trigger's polygonal structure is characterized by varying edge counts, ranging from 3 to 8. Furthermore, each individual trigger is endowed with randomly generated color and texture attributes.
>
>
> ---
>
> **[Q1] Workflow of Django**
>
> We extend our gratitude to the reviewer for highlighting the presentation issue in our paper. In response, we have introduced a comprehensive overview figure (rebuttal PDF Figure 2) during the rebuttal, which provides a visual representation of our Django framework. Django operates through two distinct stages:
>
> 1. Compromised Label Proposal via Backdoor Leakage: In this initial stage, we propose  a lightweight screening algorithm that swiftly identifies a small subset of victim-target label pairs. This selection is grounded in the observation that the behavior of a poisoned model on victim samples tends to shift towards the target label, even in the absence of the backdoor trigger itself. a.k.a backdoor leakage. Further elaboration can be found in Section 3.3 (main text, lines 223-233).
>
>
> 2. Trigger Inversion via Gaussian Focus Loss: The second stage involves trigger inversion, where each chosen label pair undergoes a precision-oriented process using our proposed Gaussian Focus Loss. This process precisely and dynamically captures a small fraction of compromised bounding boxes, assigning them larger coefficients when calculating the inversion objective function. The norm of the inverted trigger from each candidate pair serves as a determinant of the model's benignity. Additional insights are available in Section 3.2 (main text, lines 184-222).
>
> ------------
>
> **[Q3] Hyperparameter settings and sensitivity**
>
> In Section 4.2 (lines 326-332), we discuss and evaluate the sensitivity of hyperparameters in the pre-processing procedure ($h$ and $\omega$), and the main results are shown in the main text Figure 5b.  Recall the values of $h$ and  $\omega$ strike the balance between efficiency and effectiveness of the pre-processing procedure
>
> We conducted experiments by setting $h$ from 1 to 10 and $\omega$ from 0.1 to 0.8. For each combination of $h$ and $\omega$, we recorded the True Positive Rate (the ratio of selecting ground-truth label pairs) and the corresponding number of selected pairs in total for each model architecture. Figure 5b illustrates the results. We observed that different model architectures require slightly different preprocessing hyper-parameter settings to achieve the optimal trade-off between efficiency and effectiveness. However, the True Positive Rate (TPR) quickly saturates after selecting a reasonably small portion of label pairs (less than 200 out of 2602 pairs for all architectures). To provide more detailed information, we will include specific values of $h$ and $\omega$ at each changing point of Figure 5b in the revised version.
>
> We also evaluate the Django performance under different settings of IoU threshold, initialization region size and score threshold. Please find more details in Appendix D.
>
> During the rebuttal, we further evaluate the impact of two hyperparameters(initial mean $\mu$ and variance $\sigma$) in the Eq.4 on Django. We randomly sample 8 models (4 trojan and 4 benign) for each architecture that was trained on the synthesized traffic sign dataset. Besides the default values we report in the paper ($\mu$=0.1,$\sigma$=2), we set 5 more groups of initial values and report the detection performance.  As shown in rebuttal PDF Table 2, Django remains effective under different initializations.
>
> ------------

---

> > ### Comment · Reviewer_tRsV · 2023-08-13
> > **Many thanks to the authors for answering almost all of my questions and concerns.**
> >
> > I have two last questions and concerns as follows:
> >
> > Can the authors explain how the authors obtain the best model during the training process before using the best model for the evaluation phase? Please let me know if I miss something here.
> >
> > The source code has not been released, so it is quite hard to check the correctness and consistency of the implementation regarding the proposed model’s theory as well as the reproducibility of the experiments.
> >
> > I am willing to increase the score when the authors solve my above-mentioned questions and concerns.

---

> > > ### Author Response · Authors · 2023-08-14
> > > **Replies to followup questions**
> > >
> > > **[Q1] Model training and evaluation**
> > >
> > > We appreciate the reviewer for raising the subsequent questions. We have identified two potential interpretations of the term 'model' in the question and have addressed them individually:
> > >
> > > 1. If 'model' refers to the subject models (poisoned and clean object detection models) used to evaluate our Django framework, we clarify that these models were pre-trained and obtained from TrojAI rounds 10 and 13. They were trained until convergence, meeting specific criteria such as clean mAP for benign models, poison mAP, and clean mAP for poisoned models.
> > >
> > >
> > > 2. If 'model' pertains to our proposed Django framework, we emphasize that, akin to other reverse-engineering based backdoor detection methods[49, 15, 51], Django is a non-parametric method that does not necessitate a training process. When provided with a model and a small set of clean samples, Django determines whether the model is poisoned by analyzing the size of the inverted trigger. For meta-classifier based backdoor detection approaches [17, 60], which require clean and poison models for training, we present their 5-fold cross-validation outcomes in the main text's Table 2. We intend to offer further clarity on this matter.
> > >
> > > Please let us know if we have misunderstood your questions. We are open to a thorough discussion.
> > >
> > >
> > > ----
> > >
> > > **[Q2] Code availability**
> > >
> > > We promise to release all the source code to reproduce our experimental results upon publication.

---

> > > > ### Comment · Reviewer_tRsV · 2023-08-15
> > > >
> > > > Thanks to the authors for addressing my questions and concerns. I have no more questions.
> > > >
> > > > In summary, I am satisfied with some of the authors' explanations, so I will increase my initial score from 4 to 5.

---

### Official Review · Reviewer_VAnQ · 2023-07-06

**Soundness:** 3 good
**Presentation:** 3 good
**Contribution:** 3 good
**Rating:** 5
**Confidence:** 3

**Summary:**

This paper investigates the problem of detecting trojans in the context of Objection Detections. The authors first observed that the poison effect can vary significantly across bounding boxes in object detection models due to its dense prediction nature, which leads to a misalignment issue for existing trigger reverse-engineering methods. To solve this problem, they proposed the Django framework built upon the Gaussian weighting scheme to prioritize more vulnerable victim boxes. Extensive empirical results are conducted several objects, datasets, and models.

**Strengths:**

- The problem of detecting backdoors in Objective Detections is of sufficient interest to the NeurIPS community.

- The paper is well-written and easy to follow.

- The empirical evaluations over three objects, 16 detection image datasets, three model architectures, and two types of attacks are convincing.

**Weaknesses:**

- Why not consider non-trigger-inversion-based methods for detection? Regarding backdoor detection problems in the case of classification problems, many methods do not rely on inverting the trigger, e.g., the STRIP.  I am curious about the performance of applying these methods for detection.

- Do you have any theoretical comprehension of the choice of gaussian? How about some other options?


**Questions:**

- The different weighting schemes (of the poison effect) on the bounding boxes seem to resemble the observations in CNN that different magnitudes of neurons for clean and backdoor inputs. Do you have any comments on this?


**Limitations:**

Please see my comments above

---

> ### Author Rebuttal · Authors · 2023-08-10
>
> **[W1] Comparison with non reverse-engineering based backdoor detection methods**
>
> Please refer to **global response A1**.
>
> ------
>
> **[W2] Theoretical comprehension of the choice of gaussian**
>
> Please refer to **global response A2**.
>
> ------
>
> **[Q1] Correlation between weight schemes and compromised neuron magnitude**
>
> We believe our findings are consistent with the disparity in neuron magnitudes observed between clean and backdoored samples in CNN models [29]. In the context of object detection models, we suspect that the elevated activation values of target labels within compromised bounding boxes can be attributed to the pronounced magnitudes of compromised neurons within intermediate layers.

---

> > ### Comment · Reviewer_VAnQ · 2023-08-21
> >
> > Thanks to the authors for addressing my concerns. I am happy to maintain my positive rating.

---

### Official Review · Reviewer_5mmp · 2023-07-07

**Soundness:** 3 good
**Presentation:** 3 good
**Contribution:** 3 good
**Rating:** 7
**Confidence:** 4

**Summary:**

This paper proposes a novel trigger detection method that works on object detection tasks. The proposed method leverages a novel Gaussian Focus Loss to calibrate the misaligned loss during trigger inversion. The evaluations presented in the paper demonstrate that the proposed method is effective in different settings including against adaptive attack.

**Strengths:**

1. This paper has revealed and discussed a generalized phenomenon in the object detection model trigger inversion, i.e., the misalignment of CE Loss and ASR. From my perspective, this finding can shed light on future works on defending against backdoor attacks in object detection tasks and also propose a good question for the backdoor attacks in the setting.
2. The proposed method has leveraged the finding mentioned above and solved the challenges when applying such a finding, I think the proposed method has good soundness.
3. The evaluations are comprehensive and the proposed method is effective. In addition, the computational cost is quite acceptable.
4. To sum up, the proposed method is simple yet effective while the discovered phenomenon can provide new perspectives in this field.

**Weaknesses:**

The limitations of the proposed method are not discussed in this paper.


**Questions:**

Please refer to the weaknesses.

**Limitations:**

This paper has not discussed the limitations. A potential limitation of the proposed method may lie in the performance against different types of triggers (e.g., dynamic triggers).

---

> ### Author Rebuttal · Authors · 2023-08-10
>
> **[W1] No limitation discussed**
>
> As discussed in the main text (lines 107-113), our primary focus in this paper is on attacks that use static polygon triggers, which are more feasible in real-world scenarios. How to effectively inject more complex attacks such as WaNet[38], DFST[7] and dynamic attack[43] into object detection models is still an open question. We leave it to future work.  We're planning to enhance the clarity of our writing and will also include a section that highlights the limitations of our approach.

---

### Official Review · Reviewer_sH6B · 2023-07-09

**Soundness:** 3 good
**Presentation:** 3 good
**Contribution:** 3 good
**Rating:** 6
**Confidence:** 4

**Summary:**

The key idea of this paper is to propose Django, an adaptation of the trigger reverse-engineering technique used for detecting backdoored models in classification to object detection models. The paper shows that existing trigger reverse-engineering techniques are ineffective in object detection and finds that loss misalignment is the primary reason for the less effectiveness. The paper then proposes a new loss function that leverages Focal loss, a well-studied loss function in object detection, against compromised object detection models. In evaluation, Django is more effective than the baseline methods in backdoor detection with less computational overhead. The paper shows Django is effective against a few adaptive attacks.

**Strengths:**

1. The paper proposes a backdoor detection method for object detection models.
2. The paper presents why existing detection (trigger reverse-engineering) methods are ineffective.
3. Django implements a novel loss function that addresses the problems identified in 2.
4. The paper runs an evaluation against existing techniques and adaptive attacks.

I like the contribution of this paper proposing a backdoor detection method for object detection models. I don't think this is a groundbreaking contribution, but it is also not a trivial contribution. I believe Django could be a good baseline for backdoor detection in object detection.

**Weaknesses:**

1. I expect the related work could be a bit more comprehensive in backdoor attacks and detection "in object detection."
2. The evaluation mostly compares the effectiveness against trigger reverse-engineering techniques; comprehensiveness would be nice.

Related work

> Since the paper addresses backdooring in object detection, the related work should discuss the backdoor attacks and defenses in object detection. As backdooring has been studied for a while, there are many backdoor attack papers and proposed defenses. It would also be nice for readers to identify the novelty of this work much more clearly.

Evaluation

> I believe that there are multiple approaches to backdoor defenses (for example, I can think of a simple way like STRIP). Hence, I'd like to see a comprehensive categorization of backdoor defenses (perhaps in the related work section) and a comparison of Django against those missing in the current evaluation. The focus is not to show Django outperforms all the defenses but to show where Django lies in backdoor detection and how effective/ineffective Django is compared to different approaches.


**Questions:**

No question.

**Limitations:**

The paper discusses the limitations in Line 333--342.

---

> ### Author Rebuttal · Authors · 2023-08-10
>
> **[W1] More comprehensive related work in object detection**
>
> We thank the reviewer for the detailed suggestion regarding related work. We will add a new section to introduce more detail regarding backdoor attack and defense in the context of object detection.
>
> -----
>
> **[W2] Comparison with non reverse-engineering based backdoor detection methods**
>
> Please refer to **global response A1**.

---

> > ### Comment · Reviewer_sH6B · 2023-08-17
> > **Thank You**
> >
> > Thank the authors for the detailed response regarding additional defenses. It addresses my concerns.

---

### Official Review · Reviewer_TXHx · 2023-07-23

**Soundness:** 3 good
**Presentation:** 3 good
**Contribution:** 3 good
**Rating:** 6
**Confidence:** 4

**Summary:**

In this paper, the author proposes a new method for detecting backdoors in object detection models. The author finds that directly applying backdoor detection methods for classification models to object detection models results in a loss misalignment problem. Therefore, the author suggests assigning different weights to different bounding boxes during the trigger inversion optimization process. As such, they propose a new optimization loss function in the paper, Gaussian Focus Loss, to better recover triggers. Additionally, to reduce computational overhead, the author also proposes a pre-processing method to decrease the number of label pairs that need to be scanned.

**Strengths:**

1. This paper summarizes and analyzes the loss misalignment phenomenon that occurs during trigger inversion in object detection models and proposes a new optimization function based on this phenomenon. Compared to directly applying existing trigger inversion methods for classification models to object detection models, the method proposed in this paper can achieve better results at a lower cost.
2. The pre-processing method proposed in this paper, based on backdoor leakage, reduces the cost of scanning the model. This method can also be applied to other backdoor detection scenarios.
3. The experimental results in this paper support the effectiveness of the proposed method.
4. The paper is well-organized and easy to follow.

**Weaknesses:**

1. Some hyperparameters mentioned in the paper have not been carefully studied for their impact on the method's effectiveness, such as the two parameters in Eq.4 and the h in label pair pre-processing.
2. All models tested in the paper are selected from TrojanAI, but the paper does not detail the poisoned conditions of the models, such as the poisoning rate, trigger size, and trigger type.
3. There are no recovered trigger samples given in the paper.
4. Code is missing.

**Questions:**

1. Is this method effective for composite types of triggers? (e.g., "Composite Backdoor Attack for Deep Neural Network by Mixing Existing Benign Features")
2. Is the method proposed in this paper effective for other types of backdoors? (e.g. Backdoors in the physical world "Dangerous Cloaking: Natural Trigger based Backdoor Attacks on Object Detectors in the Physical World")
3. The threat model requires some clean samples; are these samples selected from training samples? If not, would this impact the inversion effectiveness?
4. The paper categorizes backdoor attacks on object detection into two types: misclassification and evasion. Does the misclassification attack include an object-appearing attack, where a bounding box that does not exist in the ground truth appears? ( "BadDet: Backdoor Attacks on Object Detection" and "Clean-image Backdoor: Attacking Multi-label Models with Poisoned Labels Only")

**Limitations:**

The author did not mention limitations in the article. Although this method can detect backdoors in object detection models, it is limited to polygon triggers, and other types of triggers have not been explored.

---

> ### Author Rebuttal · Authors · 2023-08-10
>
> **[W1] Hyperparameter settings and sensitivity**
>
> In Section 4.2 (lines 326-332), we discuss and evaluate the sensitivity of hyperparameters in the pre-processing procedure ($h$ and $\omega$), and the main results are shown in the main text Figure 5b.  Recall the values of $h$ and  $\omega$ strike the balance between efficiency and effectiveness of the pre-processing procedure
>
> We conducted experiments by setting $h$ from 1 to 10 and $\omega$ from 0.1 to 0.8. For each combination of $h$ and $\omega$, we recorded the True Positive Rate (the ratio of selecting ground-truth label pairs) and the corresponding number of selected pairs in total for each model architecture. Figure 5b illustrates the results. We observed that different model architectures require slightly different preprocessing hyper-parameter settings to achieve the optimal trade-off between efficiency and effectiveness. However, the True Positive Rate (TPR) quickly saturates after selecting a reasonably small portion of label pairs (less than 200 out of 2602 pairs for all architectures). To provide more detailed information, we will include specific values of $h$ and $\omega$ at each changing point of Figure 5b in the revised version.
>
> We also evaluate the Django performance under different settings of IoU threshold, initialization region size and score threshold. Please find more details in Appendix D.
>
> During the rebuttal, we further evaluate the impact of two hyperparameters(initial mean $\mu$ and variance $\sigma$) in the Eq.4 on Django. We randomly sample 8 models (4 trojan and 4 benign) for each architecture that was trained on the synthesized traffic sign dataset. Besides the default values we report in the paper ($\mu$=0.1,$\sigma$=2), we set 5 more groups of initial values and report the detection performance.  As shown in rebuttal PDF Table 2, Django remains effective under different initializations.
>
> ------------
>
> **[W2] Model configuration details**
>
> In Appendix A, we provide comprehensive details regarding the datasets and model architectures employed in this study. To provide further elucidation, it is important to note that the poison rate varies within the range of 0.1% to 8% across diverse models sourced from TrojAI r10 and r13. Similarly, the trigger size exhibits a range of 1x1 to 22x22, representing a scale of 0.001% to 0.7% relative to the input dimensions. Pertaining to the hyper-parameters utilized in model training, the learning rate is stochastically assigned, spanning from 1.56e-08 to 1e-04 across different models. The number of epochs for training spans from 6 to 100, while the batch size ranges from 4 to 32. As for the model performance metrics, the average clean mAP across the models attains a value of 0.7979, while the average poison mAP stands at 0.7680. We assess TrojAI models using polygon triggers, encompassing two distinct attack types: misclassification and evasion attacks. The trigger's polygonal structure is characterized by varying edge counts, ranging from 3 to 8. Furthermore, each individual trigger is endowed with randomly generated color and texture attributes.
>
> Please refer Appendix C for the detail settings of baseline methods
>
> ------------
>
> **[W3] Recovered trigger samples**
>
> We attach the inverted trigger from TrojAI round13 model ID-7 and ID-120 in rebuttal PDF Figure 1. Compared to ground truth triggers, Django inverted triggers have good visual similarity.
>
> ------------
>
> **[W4] Missing code**
>
> We will release the code upon publication.
>
> ------------
>
> **[Q1] Evaluation on composite attack**
>
> During the rebuttal phase, we conducted an evaluation of Django using 10 models that were trained on the traffic sign synthesis dataset. This set included 5 clean models and 5 trojan models poisoned by composite attack. As indicated in Table 5 of the rebuttal PDF, Django achieved an 0.9000 ROC-AUC for detecting composite attacks. It's worth noting that the composite attack does not rely on an explicit trigger. Instead, it leverages a clean object A to serve as the trigger for attacking another object B. Interestingly, Django is capable of effectively reversing this process, essentially identifying a trigger that closely mimics the pattern of object A. This ability enables Django to detect composite backdoors with a high level of accuracy.
>
> ------------
>
> **[Q2] Evaluation on Physical attack**
>
> Dangerous Cloaking uses homemade dataset with t-shirt as trigger to realize the physical backdoor attack. Unfortunately, the paper did not release the code and dataset for our reproduction. We will further contact the authors or collect samples by ourselves and evaluate the effectiveness of Django in the revision.
>
> ------------
>
> **[Q3] Source of the inversion samples**
>
> All the samples we used for Django reverse-engineering are from the validation set. To evaluate the effect of different sources, we conduct the experiments on 10 models trained on COCO dataset (5 clean and 5 poison).  For each model, we run Django twice with samples from different sources, i.e., randomly sample 10 images from each class in the training set and validation set separately. The detection performance is shown in rebuttal PDF Table 4. We can see that Django is not sensitive to the source of the clean images used for inversion. We will clarify in the revision.
>
> ------------
>
> **[Q4] Evaluation on object-appearing attack**
>
> Yes, according to our definition, the object-appearing attack is a special case of the misclassification attack with the background \mathcal{empty} class as the victim class. To demonstrate the effectiveness of Django on object-appearing attacks, we evaluate Django on 10 Baddet models and 10 Clean-image backdoor models during rebuttal. For each type of attack, we mix 5 clean models with 5 poisoned models with object-appearing triggers. The evaluation results are shown in rebuttal PDF Table 5. Django achieves 0.8 ROC-AUC on both Baddet and clean-image object appearing attacks.

---

> > ### Comment · Reviewer_TXHx · 2023-08-13
> >
> > Thank the authors for the efforts made to address my concerns. I have no more questions.

---

### Author Rebuttal · Authors · 2023-08-10

**Global Response**

We extend our sincere gratitude to all the esteemed reviewers for their invaluable and astute feedback. In the Global Response section, we shall diligently address the common questions raised and supplement the document with additional figures and tables to enhance clarity. Besides, we will reply to the specific inquiries and suggestions of each reviewer individually. “Q”, “W” and “L” indicate the question, weakness and limitation mentioned by the corresponding reviewer. e.g. “Q2@sH6B” denotes the second question brought by the reviewer sH6B. “A” denotes our answer. Tables and figures detailing the additional experiments can be found in the rebuttal PDF.

------------------------

**[W2@sH6B, W1@VAnQ] Clarification of our threat model and evaluation of other types of backdoor defenses**

**A1**:  As discussed in the threat model (main text lines 107-113), our proposed Django framework falls under the category of backdoor model detection, aligning with a line of existing works[49, 15, 51]. In this scenario, defenders are required to classify the subject model as trojaned/benign with access only to a limited set of clean samples but no poisoned samples. Model-level backdoor detection techniques are usually executed offline before model deployment.

On the contrary, the techniques mentioned by the reviewers, such as STRIP[13] and Detection Cleanse[3], belong to another type of backdoor defense known as backdoor sample detection. This defense approach operates under a completely different threat model and serves a distinct purpose. Defenders aim to discriminate trojaned/benign input samples on-the-fly, requiring access to the poison samples.  Therefore, it may not be appropriate  to compare our proposed Django with STRIP and Detection Cleanse. We will further clarify in the revision.


However, during our rebuttal, we encountered a related work FreeEagle[1] that extends STRIP to encompass trojaned model detection techniques. In line with the paper's configuration, we replicated the setup and conducted a comparative analysis between STRIP and Django. Our evaluation involved a random sampling of 8 clean models and 8 poisoned models, all trained on the traffic sign synthetic dataset. Rebuttal PDF Table 1 presents evaluation results. We can see that Django is able to achieve 0.9375 ROC-AUC while STRIP  has 0.6250. It is because STRIP is not capable of detecting label specific triggers[1]. Moreover, it's important to note that STRIP's superimposing operation has the potential to introduce additional objects onto the fused image. This is particularly significant due to the dense output nature of object detection models. Consequently, this operation may influence the entropy scores of bounding boxes, which could ultimately lead to a decrease in its overall detectability.

In addition to reverse-engineering based backdoor model detection techniques, we also compare Django with two state-of-the-art meta-classifier based techniques: MNTD[60] and MF[17]. As shown in the main text Table 2, Django outperforms the two baselines by large margins on all three datasets. Please refer to the detailed discussion from main text Line 292-313.


[1] Fu, Chong, et al. "FreeEagle: Detecting Complex Neural Trojans in Data-Free Cases." arXiv preprint arXiv:2302.14500(2023).

------------------------

**[W2@VAnQ, W2@tRsV] Intuition and theoretical comprehension of Gaussian Focal Loss**

**A2**: The selection of the Gaussian distribution stems from empirical observations made during our exploration of the underlying reasons behind the misalignment issue in evasion and misclassification attacks(main text, lines 162-173). This choice is supported by the discussions in Section 3.2 (main text, lines 199-209), where we highlight the constraints of the Focal Loss method.

Specifically, we observed the cause of misalignment is due to the unequal poisoning effect on individual bounding boxes. Only boxes with moderate confidence shall be focused dynamically through the entire inversion procedure. Therefore, naive Focal Loss is insufficient as it only focuses on boxes with low confidence, i.e. hard examples.

Furthermore, we observe our goal is highly aligned with the natural bell shape of the probability density function of gaussian distribution. Through adjusting the mean and variance during inversion, the proposed gaussian weighting scheme can dynamically capture the boxes with moderate confidence scores and assign larger coefficients to encourage the transition.

Please note that any distributions characterized by centralized peaks are suitable for our intended purpose. During the rebuttal, we attempted to substitute the Gaussian distribution with the Laplace distribution, another commonly employed probability distribution. Our experimentation involved 20 models trained on the traffic sign synthetic dataset, comprising 10 clean models and 10 models poisoned with misclassification triggers. The outcomes of these experiments are presented in rebuttal PDF Table 3. It is evident that with the Laplace focus loss, Django maintains a high level of effectiveness, achieving an ROC-AUC of 0.9, whereas baseline methods only achieve 0.7. We hypothesize that the 0.05 performance decline, in comparison to the Gaussian Focus Loss, may be attributed to the sharper peak of the Laplace distribution when contrasted with the Gaussian distribution. Within our context, a reduced number of bounding boxes exhibiting moderate confidences are allocated larger coefficients in different stages, thereby potentially diminishing the emphasis on compromised boxes. We intend to present a more comprehensive array of experiments in the revised version.

---

### Decision · Program_Chairs · 2023-09-21

**Decision:**

Accept (poster)

**Comment:**

The rebuttal addressed most of the concerns raised by the reviewers, so all reviewers are in favor of accepting this paper. Hence, the AC recommends accepting it.